# Clinical efficacy of prebiotics and glycosaminoglycans *versus* placebo In dogs with food responsive enteropathy receiving a hydrolyzed diet: A pilot study

**Barbara Glanemann**[1], **Yeon-Jung Seo**[2], **Simon L. Priestnall**[3], **Oliver A. Garden**[4], **Logan Kilburn**[5], **Mariana Rossoni-Serao**[5], **Sergi Segarra**[6], **Jonathan P. Mochel**[2], **Karin Allenspach**[7] *

1 Department of Veterinary Clinical Sciences and Services, Royal Veterinary College (RVC), University of London, London, United Kingdom, 2 Biomedical Sciences, College of Veterinary Medicine, Iowa State University, Ames, IA, United States of America, 3 Department of Pathobiology & Population Sciences, Royal Veterinary College, Hatfield, United Kingdom, 4 Clinical Sciences and Advanced Medicine, University of Pennsylvania College of Veterinary Medicine, Philadelphia, PA, United States of America, 5 Animal Sciences, Iowa State University, Ames, IA, United States of America, 6 R&D Bioiberica S.A.U., Barcelona, Spain, 7 Department of Veterinary Clinical Sciences, Iowa State University, Ames, IA, United States of America

* allek@iastate.edu

**Data Availability Statement:** All relevant data are within the manuscript and its Supporting Information files.

## Abstract

Induction of remission is easily achieved with dietary treatment in dogs diagnosed with Food Responsive Chronic Diarrhea (FRD). Administration of prebiotics and glycosaminoglycans (GAGs) may improve epithelial cell integrity and therefore be useful as adjunct treatment. This study evaluated whether the relapse rate of FRD dogs that are switched back to a normal diet can be influenced using supplemental treatment with prebiotics and GAGs. A randomized, controlled clinical trial (RCCT) was performed in dogs diagnosed with FRD. Dogs were diagnosed based on clinical exclusion diagnosis, endoscopic biopsies showing predominantly lymphoplasmacytic infiltration, and response to dietary treatment. Dogs were randomized to be fed a combination of prebiotics and GAGs (group 1) or placebo (group 2) in addition to a hydrolyzed diet. At week 10, a second endoscopy was performed and dogs were switched back to normal diet. Relapse rate was monitored every 2 weeks after that until week 18. Statistical analysis was performed for each outcome (Canine Chronic Enteropathy Clinical Activity Index (CCECAI), clinicopathological data, endoscopic scoring, mWSAVA histological scoring index (mWSAVA), and number of relapses following switch to normal diet) using a linear mixed effects model for group comparison. Time, group, and their interactions were included as a fixed effect, whereas each dog was treated as a random effect. Of the 35 dogs enrolled into the clinical trial, 10 in each group reached the point of second endoscopy. A total of 13 dogs (n = 8 in group 1 and n = 5 in group 2) reached the trial endpoint of 18 weeks. After switching back to normal diet, none of the dogs in either group relapsed. No significant differences were found over time or between groups for CCECAI, endoscopy scoring and histological scoring. Although there was a clinical worsening in the placebo group after switching back to the original diet, this was not statistically significant

**Funding:** This clinical trial study was funded by Bioiberica S.A.U., Barcelona, Spain.

**Competing interests:** No authors have competing interests.

(CCECAI p = 0.58). Post-hoc power calculation revealed that 63 dogs per group would have been needed to detect statistically significant differences in CIBDAI between treatment groups. Standard dietary treatment induced rapid clinical response in all cases, however, additional supplementation with prebiotics and GAGs did not significantly improve clinical outcome within 4 months after switching back to normal diet. Since there are very few RCCT published in CE in dogs, this pilot study provides important power analyses for planning of further studies.

## Introduction

Canine chronic enteropathies (CE) are a group of inflammatory intestinal diseases that are usually defined by response to treatment as food-responsive disease (FRD), antibiotic-responsive disease (ARD) and (idiopathic) inflammatory bowel disease (IBD), with the latter also being termed steroid-responsive disease (SRD) by most authors [1]. FRD is defined as chronic diarrhea that responds quickly (within 10–14 days) to feeding a novel protein or hydrolyzed diet exclusively. If they respond, the recommendation for these dogs is to keep them on the diet for 10–12 weeks, before switching them back to their normal diet [1, 2]. While induction of remission is easily achieved with dietary treatment in dogs with FRD, relapses can occur when the dogs are switched back to their normal diet after the trial elimination diet [2, 3]. Glycosaminoglycans (GAGs) have previously been reported to have beneficial effects on intestinal epithelial cells. In particular, increased expression of tight junction proteins, such as zonulin-1, have been observed in mice as well as human colon cancer cell lines after treatment with GAGs [4, 5]. This effect is mediated through binding of GAGs to CD44- and TLR4-receptors on the surface of intestinal epithelial cells, which results in increased intestinal stem cell proliferation, and therefore enhances intestinal regeneration [6]. The effect of GAGs on enhancing the barrier function of the intestinal epithelium, has been shown to protect animals from translocation of pathogenic bacteria and LPS into the peripheral blood [6]. In addition, GAGs have been shown to have a beneficial effect on the composition of the microbiome, mainly by reducing the number of pro-inflammatory proteobacteria, which also results in increased fecal butyrate concentrations [7, 8]. Finally, in a recent multi-center study of FRD in dogs, GAGs in combination with prebiotics had a beneficial effect on serum markers of oxidative stress over a 18-week treatment period [9]. The aim of the work presented here was therefore to assess whether the relapse rate of FRD dogs that are switched back to a normal diet can be influenced using supplemental treatment with prebiotics and GAGs. Furthermore, the effect of additional prebiotic/GAG treatment on clinical severity, biochemical parameters, endoscopic lesion scoring and histology score was evaluated.

It was assumed that dogs with FRD present with a homogeneous phenotype, mild to moderate clinical disease, and less confounding factors (like concurrent treatment) compared to other forms of canine CE and that this might make comparison between treatment groups easier and more meaningful. In addition, correlations between routine laboratory parameters, clinical severity as assessed by the canine chronic enteropathy clinical activity index (CCECAI [1] and CIBDAI [9]), histological scores as assessed by modified WSAVA (mWSAVA) scoring index [10], profiles were investigated with the goal of identifying differences between treatment groups. Since GAGs have previously been shown to be able to modify SCFA concentrations in the feces [7], we also evaluated (SCFA) concentrations in the feces of the dogs in this study.

## Materials and methods

### Conduction of the prospective double-blinded placebo-controlled clinical trial

**Ethical approvals and products used.** The clinical trial was performed after approval of the Ethics and Welfare Committee of the RVC and adhering to ASPA standards (ASPA project licence number 70/7393). Written owner consent was obtained for all study participants. The test substance consisted of 2500mg resistant starch, 300mg prebiotics (β-glucans and manna-noligosacharides (MOS)), 200mg chondroitin sulphate, 20mg glycosaminoglycans, 560mg bentonite, 400mg flavourings (hydrolysed; of pork and poultry origin) and 20mg iron oxide (treatment group 1). The placebo product consisted of 660mg bentonite, 400mg flavourings (hydrolysed; of pork and poultry origin), and 20mg iron oxide, with no active ingredients. Both the product and placebo were formulated in identical powder (treatment group 2).

**Inclusion and exclusion criteria.** Cases eligible for inclusion were those with a chronic history ($\geq$ 3 weeks) of vomiting and/ or diarrhea ± weight loss, in which the diagnosis of chronic enteropathy was confirmed based on the following: no other cause for the clinical signs identified on routine hematology, serum biochemistry (including values of trypsin-like immunoreactivity (TLI) and canine pancreatic lipase (cPLI) within the reference ranges), ACTH-stimulation test and abdominal imaging (survey radiographs and/ or abdominal ultra-sound). In addition, histopathological findings of intestinal mucosal biopsies had to be consistent with chronic enteropathy (lymphoplasmacytic, eosinophilic or mixed inflammatory infiltration with or without architectural changes). Exclusion criteria were the presence of concurrent diseases or treatment with antimicrobials and/or steroidal or non-steroidal anti-inflammatory drugs in the 7 days prior to presentation. In addition, lactating or pregnant bitches, dogs < 6 months of age and dogs < 5 kg body weight were not included in the study.

**Study design.** Dogs recruited for the clinical trial were seen at four separate visits: visit 1 (recruitment, inclusion and dispensing of trial capsules once histopathological results of intestinal biopsies were available approximately 10 days later), visit 2 (14 ±3 days after starting the trial medication), visit 3 (70+/- 2 days after starting the trial medication), and visit 7 (126 ±3 days after starting the trial medication; see also Fig 1).

The procedures performed at each visit can be seen in **Table 1**.

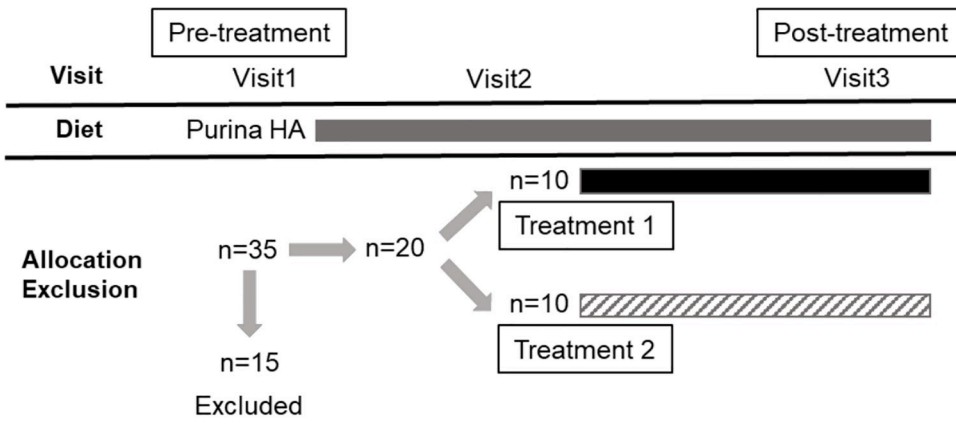

**Fig 1. Study design.** A total of thirty-five dogs were enrolled and fifteen dogs were excluded. Each box represents the period when the dogs were on Purina HA diet (grey), Treatment 1 with GAG (black), and Treatment 2 with placebo (upward diagonal).

**Table 1. Outline of procedures performed at each visit.**

| Visit Number | Physical examination (general and IBD) | CCECAI | Blood sample | Urine sample | Faecal sample (Diagnostic) | Ultra-sound | Endo-scopy |
|---|---|---|---|---|---|---|---|
| Visit 1 | ✓ | ✓ | ✓ | ✓ | ✓ | ✓ | ✓ |
| Visit 2 | ✓ | ✓ | ✓ | | | | |
| Visit 3 | ✓ | ✓ | ✓ | ✓ | | | ✓ |
| Visit 7 | ✓ | ✓ | ✓ | ✓ | | | |

After confirmation of the diagnosis, patients were randomized to receive either the product or placebo in addition to the hydrolyzed diet (Purina H/A, St. Louis, MO, USA). Patients were withdrawn from the study if their CCECAI at visit 2 or 3 increased more than 30% from the previous visit, if they suffered from an adverse event associated with the study medication or if owners could not adhere to the daily medication routine and/ or attend scheduled study visits.

## Randomization and blinding

The patients were discharged after visist 1 from the hospital once they had recovered sufficiently from anaesthesia. The owners were then told to feed the diet exclusively whilst awaiting biopsy results. Owners were contacted approximately one-week post discharge to assess response to diet. Dogs who were diet-responsive and met all other inclusion criteria were enrolled and owners told to start study medication. Dogs who had not responded to the hypo-allergenic diet were fed Purina HA for a further week, reassessed and then enrolled or excluded from the study. If the dog was not enrolled onto the study, its treatment allocation was added to the end of the randomized treatment allocation protocol (RTAP) and the treatment re-allocated. Recruitment continued until twenty dogs had been enrolled with 10 dogs in each group. Both the dogs' owners and evaluators (including clinicians and those analyzing samples and tissues) had no knowledge of the treatment group assignments. The study clinical trial nursing staff acted as dispensers and were not blinded. Randomization was performed using block randomization with an allocation ratio of one-to-one, a random number generator in excel and a block size of 4. No patients required un-blinding during the course of the study.

## Assessment of clinical severity

Clinical severity was assessed using the CCECAI or CIBDAI at all 4 visits as described previously [1].

## Assessment of relapse rate after switching back to normal diet

After visit 3, all dogs were switched back to their normal diet by gradually mixing the hydrolyzed diet with the normal diet over 1 week. The owners were called weekly between visit 3 and visit 7 in order to assess clinical signs (CIBDAI) via questionnaire. Patients were considered to have relapsed if their CCECAI at visit 4 through 7 increased more than 30% from the previous visit [11].

## Endoscopy and endoscopic scoring of lesions

Gastroduodenoscopy, ileoscopy, and colonoscopy were performed according to standard operating procedures using the available video-endoscopes appropriate for the respective dog's size. Macroscopic scoring of endoscopic findings was recorded using the World Small Animal Veterinary Association (WSAVA) approved endoscopic examination report [12]. Multiple biopsies (15–20) were taken with a single-use endoscopic biopsy forceps from both the

duodenum, ileum and colon and 10–15 biopsies from each site were transferred immediately into 2% neutral-buffered formalin.

## Histopathology

Initial histopathological assessment was performed at the diagnostic pathology department of the RVC as part of the routine diagnostic procedure. However, after finalizing the study, all slides were reviewed by one of the authors (SLP), when mWSAVA standardized classification of all slides was performed [10, 12]. At this stage, SLP was blinded to the original diagnosis, the visit number and treatment group of the animals. mWSAVA scores [10] were recorded as total scores (summation of scores for duodenum, ileum and colon; $mWSAVA_{Total}$) per dog and time point, as well as site-specific mWSAVA sub-scores for fundic and antral stomach scores (fundus: $SF_{SUBTOTAL}$; antrum: $SA_{SUBTOTAL}$), duodenum ($D_{SUBTOTAL}$), ileum ($I_{SUBTOTAL}$), and colon ($C_{SUBTOTAL}$), per dog and time point.

## Analysis of fecal Short Chain Fatty Acid (SCFA) concentrations

Fecal SCFA concentrations were determined using gas chromatography (3800 Varian GC, Agilent Technologies, Santa Clara, CA). For each sample, 2.5 mL of distilled water was added to one gram of thawed feces in a conical tube and vortexed. Samples were centrifuged at 4˚C for 10 min at $2,000 \times g$ for supernatant removal. One mL of supernatant was then transferred into a 1.5 mL centrifuge tube and mixed with 0.2 mL of metaphosphoric acid for deproteinizing and 0.1 mL of isocaproic acid as an internal standard (48.3 mM; Sigma-Aldrich, Saint Louis, MO). A standard curve was generated using five concentrations of acetate, butyrate, propionate, valerate, isovalerate, and isobutyrate (Sigma-Aldrich, Saint Louis, MO). The tubes were then centrifuged at $12,000 \times g$ at 4˚C for 25 min. Aliquots of the supernatant (1 mL) from standard and fecal samples were transferred to 1.5 mL GC vials and duplicate injections of 100 uL were used by the GC for analysis. A flame ionization detector was used with an oven temperature of 60–200˚C. The Nukol capillary column (15 m x 0.25 mm x 0.25 μm; Sigma-Aldrich, Bellefonte, PA) was operated with highly purified He, as the carrier gas, at 1 mL/min. Concentrations of acetate, butyrate, propionate, valerate, isovalerate, and isobutyrate were calculated using the ratio of the peak area of each compound to the internal standard and standard curve regression. Molar proportions of SCFA (%) were calculated as the individual SCFA / total SCFA concentration $\times 100$.

## Statistical analyses

Statistical analysis was performed using R software version 3.5.1 (R Foundation for Statistical Computing, Vienna, Austria). Normality was tested for each variable using Shapiro-Wilk for all parametric approaches. When the assumption did not hold, the nonparametric test (Mann-Whitney) was used for the variables measured only at one (visit = 1) or two time points (visits = 1,3) to compare differences between two groups. With variables involving multiple time points (more than two), each outcome was analyzed using a linear mixed effects model for group comparison. For each analysis, time, group, and their interactions were included as a fixed effect, whereas each dog was treated as a random effect. To test the correlation between variables of interest, spearman's rank correlation coefficients and the corresponding p-values were calculated. For all statistical analyses, a p-value < 0.05 was considered significant.

## Results

All raw data pertaining to clinical data of the study can be found in **S2 File**.

**Table 2. Demographics of dogs included into the study.**

| Group | Age in months (at Visit 1) | Gender | Breed |
|---|---|---|---|
| 1 | 25 | FN | Lurcher |
| 1 | 44 | FN | Cross Breed |
| 1 | 56 | FE | Bearded Collie |
| 1 | 81 | FN | Staffordshire Bull Terrier |
| 1 | 120 | FN | Cocker Spaniel |
| 1 | 14 | ME | Cockapoo |
| 1 | 29 | ME | Border Collie |
| 1 | 53 | MN | Labrador Retriever |
| 1 | 91 | MN | Labrador Retriever |
| 1 | 131 | MN | Staffordshire Bull Terrier |
| 2 | 19 | FE | Whippet |
| 2 | 54 | FE | German Shepherd |
| 2 | 22 | ME | Boxer |
| 2 | 23 | ME | Border Collie |
| 2 | 23 | MN | French Bulldog |
| 2 | 36 | ME | Collie X |
| 2 | 36 | MN | Labradoodle |
| 2 | 58 | MN | Lhasa Apso |
| 2 | 73 | MN | Staffordshire Bull Terrier |
| 2 | 80 | ME | German Shepherd |

## Animals

Of the 35 dogs enrolled into the clinical trial, 10 in each group reached the point of second endoscopy (visit 3). A total of 13 dogs (n = 8 in the group 1 and n = 5 in group 2) reached the trial endpoint of 18 weeks (visit 7). The demographics of the dogs are shown in Table 2. There were no differences in median age (group 1: 64.40 months ± 40.10 (57.13 ± 36.92); group 2: 42.40 months ± 22.43 (35.00 ± 14.53); p = 0.7) breed, or sex between the treatment groups.

Six/35 dogs did not respond to the dietary treatment and were not included in the trial. Of the remaining 29 dogs, 9 dogs discontinued the study for the following reasons: Five dogs were withdrawn because of worsening of clinical signs (one dog prior to visit 2, and 4 dogs prior to visit 3). In addition, one dog was withdrawn because the dog refused to take the test compound, and one dog was withdrawn prior to visit 3 because it required treatment with antibiotics for a skin condition. Two further dogs were excluded from the trial: One dog was diagnosed with adenocarcinoma after enrollment, and one dog was withdrawn because of owner non-compliance.

## Clinical severity and biochemical parameters

All dogs responded quickly and sustained to dietary treatment for 10 weeks. There were no significant differences in CCECAI or CIBDAI between treatment groups from visit 1 to 7 (mean ± SD CCECAI for group 1: visit 1: 4.90 ± 1.52, visit 3: 2.25 ± 2.19, visit 7: 1.38 ± 0.74; for group 2: visit 1: 6.00 ± 2.94, visit 3: 2.2 ± 1.1, visit 7: 1.8 ± 2.49; mean ±SD CIBDAI for group 1: visit 1: 5.00 1.63, visit 3: 2.25 ±2.19, visit 7: 1.12 ± 035; for group 2: visit 1: 5.6 ±2.99, visit 3: 2.00 ± 1.41, visit 7: 1.8 ±2.49). Although there was a slight clinical worsening in the placebo group after switching back to the original diet (visit 3 to visit 7), this was not statistically significant (CCECAI p = 0.79; CIBDAI p = 0.65) (see Fig 2).

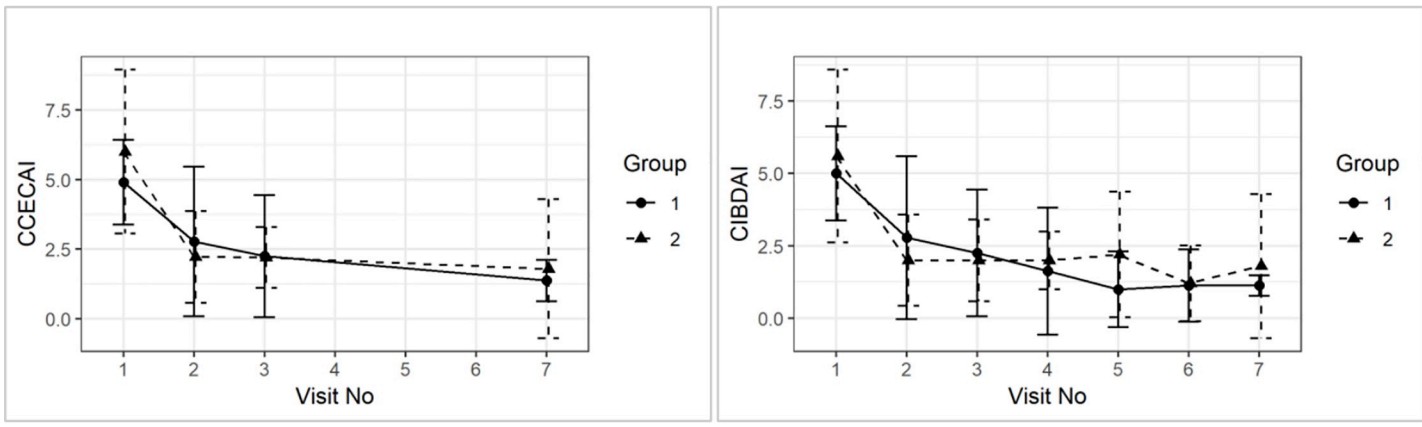

**Fig 2. Group comparison of average (±sd) CCECAI and CIBDAI over time.**

None of the dogs had serum albumin, globulin, cobalamin and folate concentrations outside of the references ranges at any time point throughout the study. Serum TLI and PLI were only measured once at visit 1 and were all within reference ranges. Mean serum albumin and globulin concentrations increased from visit 1 to visit 3 and then again to visit 7, although this was not statistically significant (mean ± SD serum albumin concentrations for group 1: visit 1: 29 ± 4.19, visit 3: 33.25 ± 3.35, visit 7: 33.79 ± 3.98; group 2: visit 1: 30.32 ±6.41, visit 3: 32.52 ±1.98, visit 7: 33.66 ±1.02; mean ± SD serum globulin concentration for group 1: visit 1: 22.12 ± 3.39, visit 3: 23. 8±3.58, visit 7: 23.86 ±4.66; for group 2: visit 1: 21.13 ±3.41, visit 3: 23.8 ±3.58, visit 7: 23.86 ±4.66; for group 2: visit 1: 21.12 ±3.41, visit 3: 23.9 ±3.04, visit 7: 26.02 ±4.55). Serum folate concentration increased from visit 1 to visit 3, and then decreased again slightly at visit 7 (serum folate concentration ± SD for group 1: visit 1: 11.22 ± 6.35, visit 3: 18.79 ± 3.62, visit 7: 14.64 ± 3.43; for group 2: visit 1: 9.85 ±2.07, visit 3: 16.54 ±4.19, visit 7: 11.16 ±5.03). Furthermore, there were no group effects on average for serum albumin ($p = 0.26$), cobalamin ($p = 0.71$), folate ($p = 0.35$), and globulin concentration over time ($p = 0.86$) (See Fig 3).

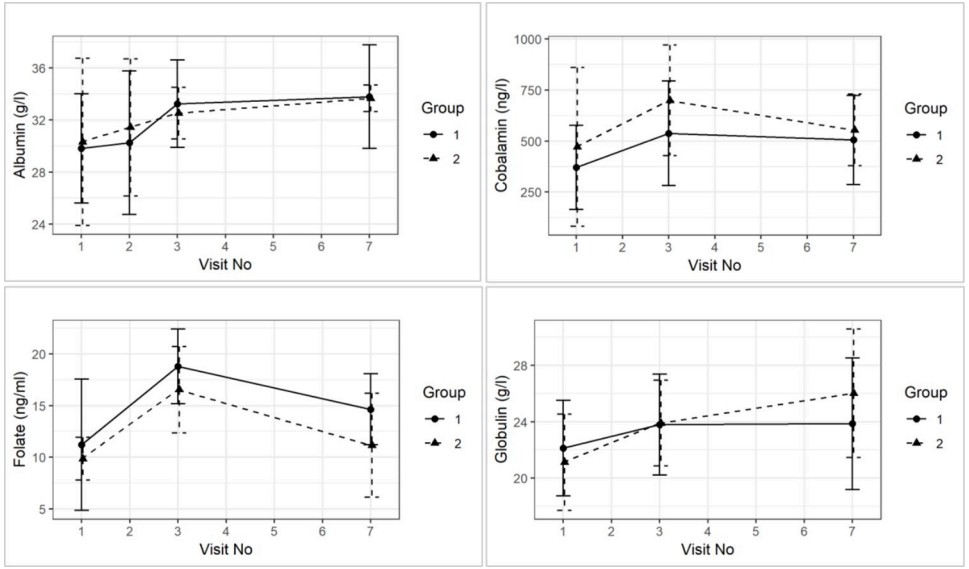

**Fig 3. Group comparison of average (±sd) for serum concentrations of albumin, cobalamin, folate and globulin over time.**

## Endoscopy scores

Endoscopy scores in the duodenum and ileum were higher than those in the colon at diagnosis (Endoscopy score D Group 1: Mean = 8.0, Median = 9.0 at visit 1; Mean = 6.4, Median = 5.0 at visit 3; Group 2: Mean = 6.3, Median = 3.5 at visit 1; Mean = 2.2, Median = 3.0 at visit 3. Endoscopy score I group 1: Mean = 8.0, Median Mean = 1.1, Median = 0.0 at visit 1; Mean = 0.5, Median = 0.0 at visit 3; Mean = 1.0, Median = 0.0 at visit 1; Mean = 0.2, Median = 0.0 at visit 3).

Although all scores reduced numerically from visit 1 to visit 3, this was not statistically significant over time. There were also no significant time and group interactions in endoscopy scores for the duodenum (p = 0.38), ileum (p = 0.71), and colon (p = 0.75) (see Fig 4).

## Modified WSAVA histology scores

There were no differences between mWSAVA$_{Total}$, nor D$_{SUBTOTAL}$, I$_{SUBTOTAL}$ and C$_{SUBTOTAL}$ at the time of diagnosis. Similarly, no statistically significant differences between groups were detected in mWSAVA$_{Total}$ (p = 0.40), nor D$_{SUBTOTAL}$ (p = 0.16), I$_{SUBTOTAL}$ (p = 0.92) and C$_{SUBTOTAL}$ (p = 0.47) over time (see Fig 5).

## Correlation analyses

A graphical display of a *correlation matrix* is given in Fig 6 for the variables CCECAI, CIBDAI, mWSAVA SF, SA, as well as mWSAVA D$_{SUBTOTAL}$, I$_{SUBTOTAL}$, and C$_{SUBTOTAL}$ scores.

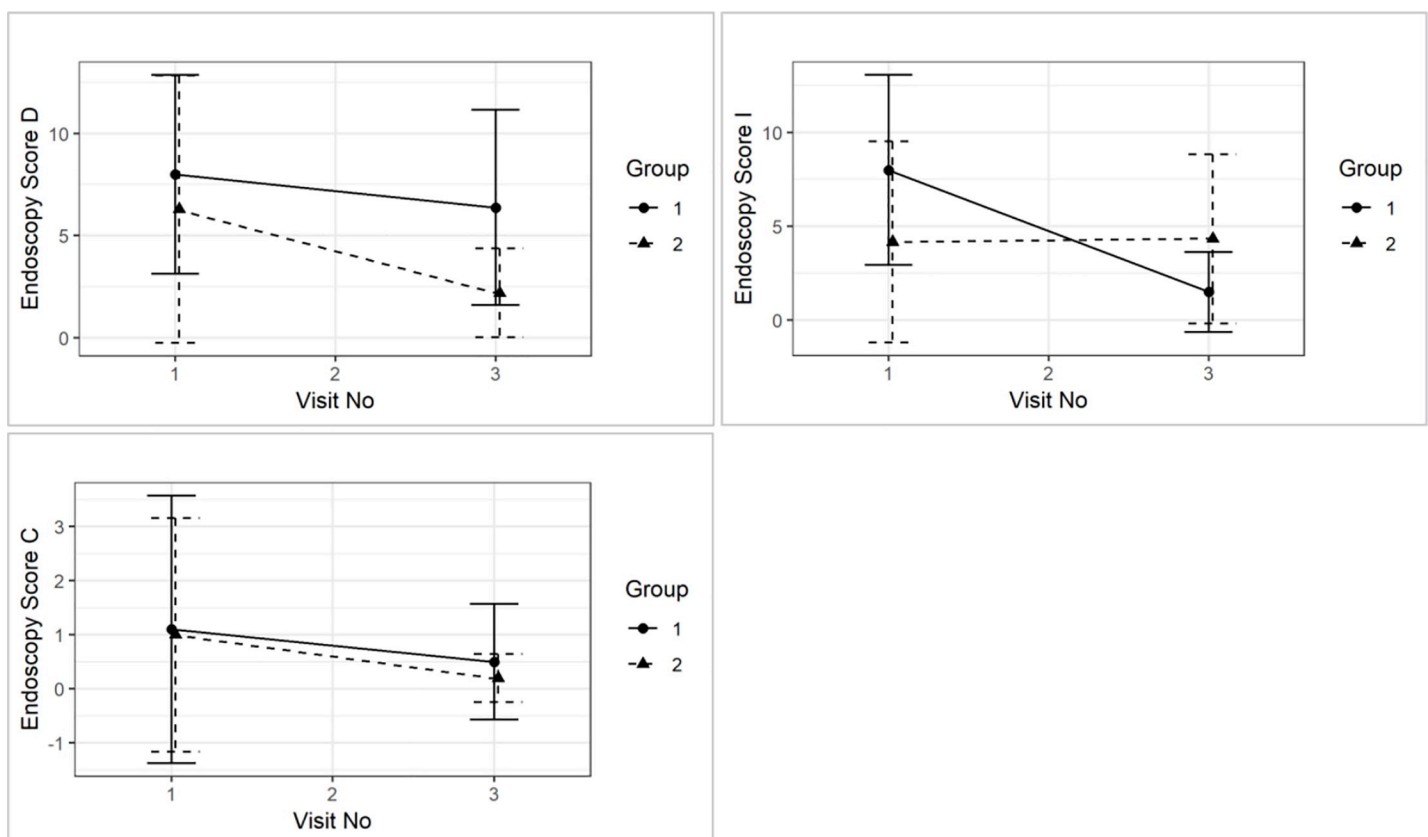

**Fig 4. Group comparison of mean (±sd) endoscopy score for duodenum (D), ileum (I), and colon (C) between visit 1 and visit 3 and between treatment groups.**

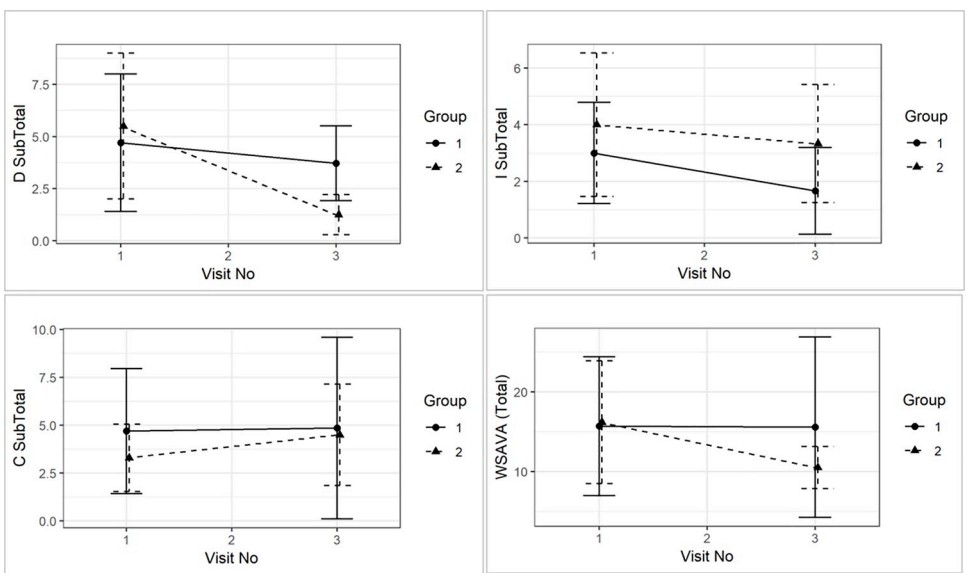

**Fig 5. Comparison of mWSAVA$_{Total}$ (p = 0.8), nor D$_{SUBTOTAL}$ (p = 0.2), I$_{SUBTOTAL}$ (p = 01.0) and C$_{SUBTOTAL}$ (p = 0.6) between visit 1 and visit 3 and between treatment groups.**

Modified mWSAVA$_{Total}$, as well as endoscopic scores for duodenum, ileum and colon, and serum albumin and cobalamin concentrations.

There were significant positive correlations between mWSAVA D$_{SUBTOTAL}$ and mWSAVA I$_{SUBTOTAL}$ scores, as well as between duodenal and ileal endoscopy scores. Furthermore, there was a significant positive correlation found between serum albumin and serum cobalamin concentrations within individual dogs.

## Post hoc power analysis

A post-hoc sample size calculation was performed based on the slope of CCECAI and CIBDAI scores between groups using visits 2 and 7, in ordert o detrmine the sample size for detecting a difference in relapse rates between the groups after switching back to normal diet. Based on CCECAI scores, the sample size required to detect a significant difference between groups would have been 238 per group for each time point. Based on CIBDAI scores, the sample size required would have been 63 per group (for each time point).

## Fecal SCFA concentrations

Fecal SCFA concentrations were determined at visits 1, 2, 3 and 7 in 14 of the dogs, 9 of which dogs were in group 1 and 5 of which were in group 2. No statistically significant differences were found in fecal concentrations of acetate, butyrate, propionate, valerate, isovalerate, and isobutyrate fecal concentrations over time and between treatment groups (see Fig 7).

## Discussion

The objective of the current study was to test the *in-vivo* clinical effects of the oral administration of GAGs and prebiotics as a supplemental treatment to hydrolyzed diet in dogs with FRD on the relapse rate of FRD dogs after switching back to normal diet.

Most dogs were medium sized breeds with a mean age of around 4.4 years, which is concordant with other recent studies on FRD dogs [2]. The severity of clinical signs was mild to

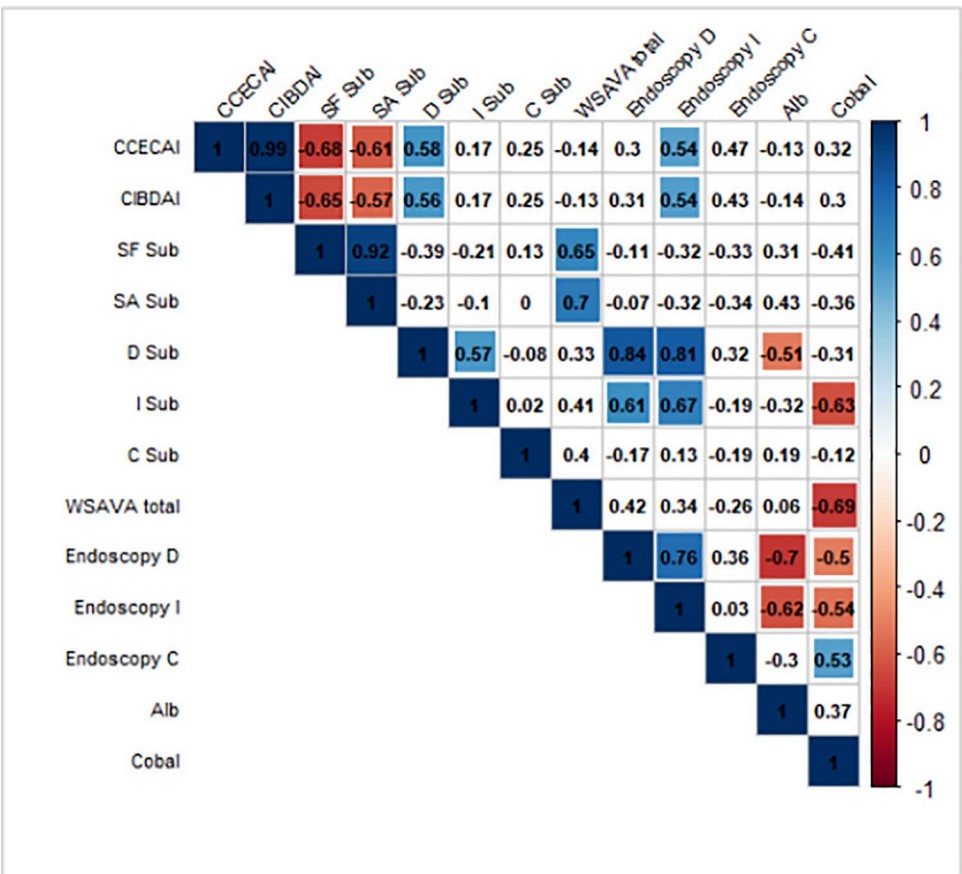

**Fig 6. Pearson correlation coefficients (combining all the visits and groups 1 and 2) using *corrplot* package in R.** Positive correlations are displayed in blue color and negative correlations are in red. Additional colors (corresponding to their correlation values) are added in the plot when the test for linear association between paired variables results in p-value < 0.05 (p-values are not shown in the plot). CCECAI = Canine Chronic Enteropathy Clinical Activity Index; CIBAI = Canine IBD Activity Index; SF sub = Stomach fundic mWSAVA subscore; SA = Stomach antral mWSAVA subscore; D sub = Duodenum subscore mWSAVA; I sub = Ileum subscore mWSAVA; C sub = Colon subscore WSAVA; mWSAVA total = combined mWSAVA subscores for all intestinal sites; Endoscopy D = Endoscopy subscore duodenum; Endoscopy I = Endoscopy subscore Iluem; Endoscopy C = Endoscopy subscore Colon; Alb = serum albumin ocnentration; Cobal = serum cobalamin concentration.

moderate and reduced significantly at visit 2 and 3, regardless of product or placebo treatment, which confirms the diagnosis of FRD. After switching back to the normal diet at visit 3 and until visit 7, none of the dogs experienced any clinical relapse in either group. Therefore, a meaningful assessment of whether the adjunct treatment had an effect on the relapse rate in these dogs cannot be made from this sample population.

No changes in biochemical parameters were noted between the groups over time, nor for endoscopic scores or histological scores, which is consistent with previous reports in FRD dogs [1, 13–15]. We did find a strong correlation between serum albumin and serum cobalamin concentration in the individual dogs, which also concurs with previous publications [1]. Furthermore, endoscopic and histological scores did not correlate with clinical scores in the dogs over time, which is a fact that has also been noted in previous studies [1, 9, 16].

However, we found strong positive correlations between duodenal and ileal endoscopic scores, as well as mWSAVA sub-scores for duodenum and ileum. This is a novel finding in our study and may represent the notion that FRD dogs present with diffuse small intestinal

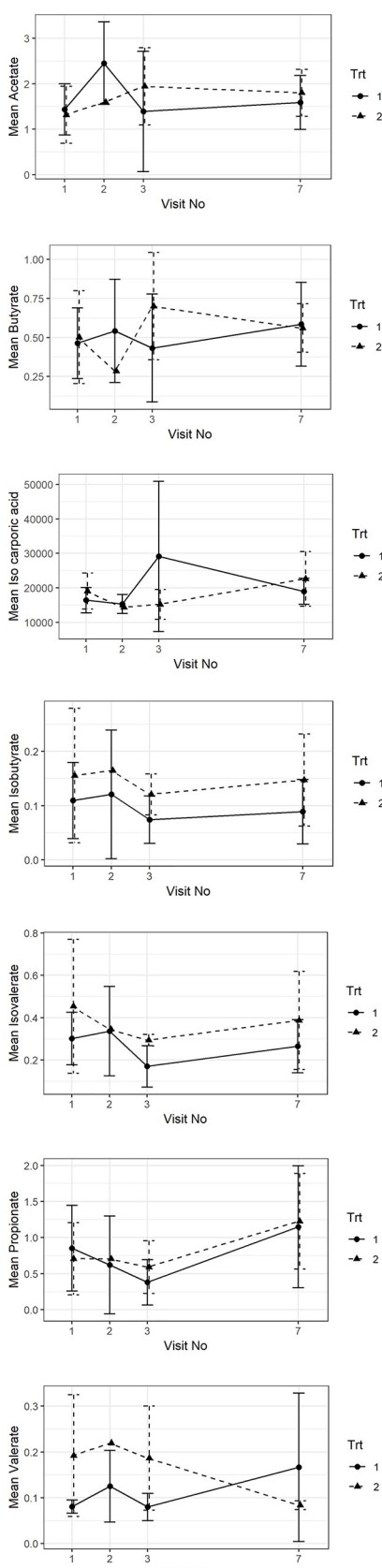

**Fig 7. Fecal SCFA (acetate, butyrate, propionate, valerate, isovalerate, and isobutyrate) concentrations over time and between groups.** Raw data from the fecal SCFA analysis can be found in S1 File.

inflammation with similar severity in the duodenum and ileum. In a recent study, it was shown that sub-scores of mWSAVA for each intestinal site (duodenum, ileum and colon) correlate better with clinical activity of disease than the total modified mWSAVA score including sub-scores of all sites [10]. Furthermore, endoscopic subscores for the duodenum and ileum in this study were higher than those for the colon at the time of diagnosis. This may represent the fact that most dogs in this study presented with predominantly small intestinal signs, and/or may be due to the fact that endoscopy scores in the small intestine seem to correlate better with clinical activity scores [1, 15].

Fecal SCFA did not show any significant changes over time or between groups in this study. It is possible that the effect of the diet was substantial and therefore, a minor effect of the supplement might have been missed.

Unfortunately, even though a substantial number of potential cases for the clinical trial were screened (n = 35), only 20 dogs (10 of each group) were enrolled, and only 13 finished the study. This resulted in the study being underpowered, with statistically significant differences in CIBDAI scores between groups expected at a minimum number of n = 63 per group. Nevertheless, this study represents one of the few published double-blinded RCCT and will be informative for the design of future clinical trials in canine FRD.

## Conclusion

Standard dietary treatment induced rapid clinical response in all cases. Because the study was underpowered, it was not possible to determine whether or not supplementation with prebiotic and GAG had an additional effect on clinical outcomes or frequency of relapses after switching back to normal diet.

## Supporting information

**S1 File. This file contains all clinical raw data fort he trial.**
(PDF)

**S2 File. This file contains all statistical output data.**
(PDF)

**S3 File. This file contains all statistical output for the analyses of fecal Short Chain Fatty Acids concentrations.**
(PDF)

## Acknowledgments

The authors would like to thank the owners of the dogs participating in this study for taking part; as well as the help of the staff of the Queen Mother Hospital for Animals (QMHA) and the Clinical Investigation Centre of the Royal Veterinary College (RVC) for their support in recruiting cases and acquiring all necessary samples.

The work presented in this manuscript has been performed at the Royal Veterinary College, London, UK.

This paper was presented as an abstract at SEVC 2019, Spain.

## Author Contributions

**Conceptualization:** Oliver A. Garden, Sergi Segarra, Karin Allenspach.

**Data curation:** Jonathan P. Mochel, Karin Allenspach.

**Formal analysis:** Yeon-Jung Seo, Simon L. Priestnall, Logan Kilburn, Mariana Rossoni-Serao, Jonathan P. Mochel, Karin Allenspach.

**Funding acquisition:** Oliver A. Garden, Sergi Segarra, Karin Allenspach.

**Investigation:** Barbara Glanemann, Oliver A. Garden, Jonathan P. Mochel, Karin Allenspach.

**Methodology:** Logan Kilburn, Mariana Rossoni-Serao, Jonathan P. Mochel, Karin Allenspach.

**Project administration:** Barbara Glanemann, Oliver A. Garden, Karin Allenspach.

**Resources:** Simon L. Priestnall, Logan Kilburn, Mariana Rossoni-Serao, Karin Allenspach.

**Supervision:** Barbara Glanemann, Oliver A. Garden, Karin Allenspach.

**Validation:** Yeon-Jung Seo, Karin Allenspach.

**Visualization:** Yeon-Jung Seo, Karin Allenspach.

**Writing – original draft:** Karin Allenspach.

**Writing – review & editing:** Barbara Glanemann, Oliver A. Garden, Logan Kilburn, Mariana Rossoni-Serao, Sergi Segarra, Jonathan P. Mochel.

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
