## [Decision Letter · Decision Letter 0]

24 Jun 2021

PONE-D-21-11631

Clinical Efficacy of Prebiotics and Glycosaminoglycans versus Placebo In Dogs with Food Responsive Enteropathy Receiving a Hydrolyzed Diet: A Pilot Study

PLOS ONE

Dear Dr. Allenspach,

Thank you for submitting your manuscript to PLOS ONE. After careful consideration, we feel that it has merit but does not fully meet PLOS ONE’s publication criteria as it currently stands. Therefore, we invite you to submit a revised version of the manuscript that addresses the points raised during the review process.

PLOS ONE welcomes primary research that contributes to the base of scientific knowledge, including negative or null results. The important point is that the data presented in the manuscript must support the conclusions drawn and the authors should avoid overstating their conclusions. We believe that this is the case for your manuscript. Therefore, we do not request you to follow the first major comment of the reviewer 2.

Please submit your revised manuscript  within 2 months. If you will need more time than this to complete your revisions, please reply to this message or contact the journal office at plosone@plos.org. Please include the following items when submitting your revised manuscript:

We look forward to receiving your revised manuscript.

Kind regards,

Mathilde Body-Malapel

Academic Editor

PLOS ONE

Journal Requirements:

2. In your Methods section, please provide additional details regarding participant consent from the owners of the animals. In the ethics statement in the Methods and online submission information, please ensure that you have specified (1) whether consent was informed and (2) what type you obtained (for instance, written or verbal). If the need for consent was waived by the ethics committee, please include this information.

This clinical trial study was funded by Bioiberica S.A.U., Barcelona, Spain.

This clinical trial study was funded by Bioiberica S.A.U., Barcelona, Spain.

5. Please note that in order to use the direct billing option the corresponding author must be affiliated with the chosen institute. Please either amend your manuscript to change the affiliation or corresponding author, or email us at plosone@plos.org with a request to remove this option.

Additional Editor Comments:

Reviewers' comments:

Reviewer's Responses to Questions

**Comments to the Author**

1. Is the manuscript technically sound, and do the data support the conclusions?

Reviewer #1: Yes

Reviewer #2: No

2. Has the statistical analysis been performed appropriately and rigorously? 

Reviewer #1: Yes

Reviewer #2: Yes

3. Have the authors made all data underlying the findings in their manuscript fully available?

Reviewer #1: Yes

Reviewer #2: Yes

4. Is the manuscript presented in an intelligible fashion and written in standard English?

Reviewer #1: Yes

Reviewer #2: Yes

5. Review Comments to the Author

Reviewer #1: Abstract- was it number of relapses that was evaluated? or was it simply whether or not they relapsed?

Introduction-

-reference missing for "relapses can occur when dogs are switched back to their normal diet after the trial elimination diet - text says (Ref)

-The statement " Therefore, supplemental treatments in addition to the diet may help reduce the number of relapses in these dogs after switching them back to their normal diet" seems out of place. It would make more sense to make this argument AFTER you discuss the benefits of GAGs on intestinal epithelial cells, etc. Suggesting moving this statement to directly after ".. oxidative stress over a 18-week treatment period.."

-... "have been observed in mice as well as human colon cancer cell lines after treatment with GAGs (reference typo - "34, 3).

-End of introduction- I think it would be a good idea to present to the reader at least some brief information as to WHY you wanted to compare fecal short chain fatty acids and the other parameters between groups.

Materials & Methods-

Endoscopy and endoscopic scoring lesions- says multiple biopsies were taken from duodenum and colon, but no mention of ileum?

Page 9- typo - "visit"

-I'm a little confused about enrollment because in the abstract it says 35 dogs were enrolled but here on page 9 it says recruitment continued until twenty dogs had been enrolled, 10 dogs in each group

Results-

Page 13- I would suggest making a flow diagram that outlines- how many dogs were screened, how many dogs initially enrolled, how many dogs completed the full study and include reasons for patient drop out

Page 16- typo-- "order to"

Discussion- well written and not overstated

Figures- adequate and appropriate

Reviewer #2: This pilot study aimed to clarify the effect of supplemental treatment with prebiotics and glycosaminoglycans (GAGs) in the relapse rate of food responsive disease (FRD) in dogs that were switched back to a normal diet in a randomized, controlled clinical trial (RCCT).

Major comments:

1. Because there were no significant differences in clinical outcomes between supplemental treatment with prebiotics and GAGs (group 1) and placebo (group 2), it is impossible to conclude that the effect of prebiotics and GAGs in the relapse rate of FRD. Without conclusions, this study would not provide any scientific significance in this field. To determine the effect of prebiotics and GAGs, this study must be conducted by using a larger sample size, as suggested by the authors.

2. This pilot study did not find any significant effects of prebiotics and GAGs on the relapse rate and fecal SCFA profiles in FRD dogs. Thus, it is unclear whether prebiotics and GAGs really improved intestinal environment in the GI tract of dogs. According to the introduction, GAGs have beneficial effects on intestinal barrier functions, microbiome, and fecal short chain fatty acid (SCFA). The authors, therefore, must analyze expression of tight junction proteins in the intestinal epithelial cells and gut microbiome in dogs that received prebiotics and GAGs or placebo.

3. Page 7, Inclusion and exclusion criteria:

To justify the diagnosis and ensure the evenness of group 1 and 2, please provide hematology and serum biochemistry data including TLI, cPLI, and ACTH-stimulation test at visit 1 and compare them between two groups.

4. Page 10, Assessment of relapse rate after switching back to normal diet: “Patients were considered to have relapsed if their CCECAI at visit 4 through 7 increased more than 30% from the previous visit.”

Please provide the scientific evidence for the relapse definition used in this study.

Minor comments:

1. Page 1: FRD is not the abbreviation of “Food Responsive Chronic Enteropathy”.

2. Page 5: Please check the references in the following sentences:

(a) While induction of remission is easily achieved with dietary treatment in dogs with FRD, relapses can occur when the dogs are switched back to their normal diet after the trial elimination diet (Ref).

(b) The effect of GAGs on enhancing the barrier function of the intestinal epithelium, has been shown to protect animals from translocation of pathogenic bacteria and LPS into the peripheral blood6,7,6,8.

3. The authors cited 15 references, but I can find reference no. more than 15 in the text. Please check the manuscript thoroughly.

4. Page 13-14, Clinical severity and biochemical parameters:

Please describe reference ranges of biochemical parameters in the text.

6. PLOS authors have the option to publish the peer review history of their article (what does this mean?). If published, this will include your full peer review and any attached files.

Reviewer #1: No

Reviewer #2: No

---

## [Author Response · Author response to Decision Letter 0]

25 Aug 2021

Rebuttal letter to editor and reviewers of PONE-D-21-11631:

Clinical Efficacy of Prebiotics and Glycosaminoglycans versus Placebo In Dogs with Food Responsive Enteropathy Receiving a Hydrolyzed Diet: A Pilot Study

Many thanks for this comment- this has been done.

2. In your Methods section, please provide additional details regarding participant consent from the owners of the animals. In the ethics statement in the Methods and online submission information, please ensure that you have specified (1) whether consent was informed and (2) what type you obtained (for instance, written or verbal). If the need for consent was waived by the ethics committee, please include this information.

Many thanks for this comment- this has been added to the section: Line 133: Written owner consent was obtained for all study participants.

This clinical trial study was funded by Bioiberica S.A.U., Barcelona, Spain.

This clinical trial study was funded by Bioiberica S.A.U., Barcelona, Spain.

Many thanks for this comment. We have taken out the funding statement in the acknowledgement section and have now updated the Funding Statement.

Many thanks for your comments. We have included all raw data and statistics data file as supplemental material.

5. Please note that in order to use the direct billing option the corresponding author must be affiliated with the chosen institute. Please either amend your manuscript to change the affiliation or corresponding author, or email us at plosone@plos.org with a request to remove this option.

Many thanks for this comment- this has been corrected. 

Additional Editor Comments:

Many thanks for this comment. We have gone through the manuscript and corrected all of the references.

Reviewer #1: 

Abstract- was it number of relapses that was evaluated? or was it simply whether or not they relapsed?

Many thanks for this comment. Since none of the dogs in this study relapsed after being switched back to a normal diet, no statistics were performed on this.

Introduction-

-reference missing for "relapses can occur when dogs are switched back to their normal diet after the trial elimination diet - text says (Ref)

Many thanks for this comment – this has been added.

-The statement " Therefore, supplemental treatments in addition to the diet may help reduce the number of relapses in these dogs after switching them back to their normal diet" seems out of place. It would make more sense to make this argument AFTER you discuss the benefits of GAGs on intestinal epithelial cells, etc. Suggesting moving this statement to directly after ".. oxidative stress over a 18-week treatment period.."

Many thanks for this comment- this sentence has now been moved further down in the paragraph:

Line 101: The aim of the work presented here was therefore to assess whether the relapse rate of FRD dogs that are switched back to a normal diet can be influenced using supplemental treatment with prebiotics and GAGs.

-... "have been observed in mice as well as human colon cancer cell lines after treatment with GAGs (reference typo - "34, 3).

Many thanks for this comment- this has been corrected.

-End of introduction- I think it would be a good idea to present to the reader at least some brief information as to WHY you wanted to compare fecal short chain fatty acids and the other parameters between groups.

Many thanks for this comment. We have added the following sentence in line 114: Since GAGs have previously been shown to be able to modify SCFA concentrations in the feces, we also evaluated (SCFA) concentrations in the feces of the dogs in this study.

Materials & Methods-

Endoscopy and endoscopic scoring lesions- says multiple biopsies were taken from duodenum and colon, but no mention of ileum?

Many thanks for this comment- this has been corrected.

Page 9- typo - "visit"

Many thanks for this comment- this has been corrected.

-I'm a little confused about enrollment because in the abstract it says 35 dogs were enrolled but here on page 9 it says recruitment continued until twenty dogs had been enrolled, 10 dogs in each group

Many thanks for this comment. We have included more detail on the recruitment and drop-outs form the study: Line 269:

Six/35 dogs did not respond to the dietary treatment and were not included in the trial. Of the remaining 29 dogs, 9 dogs discontinued the study for the following reasons: Five dogs were withdrawn because of worsening of clinical signs (one dog prior to visit 2, and 4 dogs prior to visit 3). In addition, one dog was withdrawn because the dog refused to take the test compound, and one dog was withdrawn prior to visit 3 because it required treatment with antibiotics for a skin condition. Two further dogs were excluded from the trial: One dog was diagnosed with adenocarcinoma after enrollment, and one dog was withdrawn because of owner non-compliance. 

Results-

Page 13- I would suggest making a flow diagram that outlines- how many dogs were screened, how many dogs initially enrolled, how many dogs completed the full study and include reasons for patient drop out

Many thanks for this comment. We have included more detail on the recruitment and drop-outs form the study: Line 269:

Six/35 dogs did not respond to the dietary treatment and were not included in the trial. Of the remaining 29 dogs, 9 dogs discontinued the study for the following reasons: Five dogs were withdrawn because of worsening of clinical signs (one dog prior to visit 2, and 4 dogs prior to visit 3). In addition, one dog was withdrawn because the dog refused to take the test compound, and one dog was withdrawn prior to visit 3 because it required treatment with antibiotics for a skin condition. Two further dogs were excluded from the trial: One dog was diagnosed with adenocarcinoma after enrollment, and one dog was withdrawn because of owner non-compliance. 

Page 16- typo-- "order to"

Many thanks for this comment- this has been corrected.

Discussion- well written and not overstated

Figures- adequate and appropriate

Reviewer #2: This pilot study aimed to clarify the effect of supplemental treatment with prebiotics and glycosaminoglycans (GAGs) in the relapse rate of food responsive disease (FRD) in dogs that were switched back to a normal diet in a randomized, controlled clinical trial (RCCT).

Major comments:

1. Because there were no significant differences in clinical outcomes between supplemental treatment with prebiotics and GAGs (group 1) and placebo (group 2), it is impossible to conclude that the effect of prebiotics and GAGs in the relapse rate of FRD. Without conclusions, this study would not provide any scientific significance in this field. To determine the effect of prebiotics and GAGs, this study must be conducted by using a larger sample size, as suggested by the authors.

Many thanks for this comment. As suggested by editors, we have not replied to this comment.

2. This pilot study did not find any significant effects of prebiotics and GAGs on the relapse rate and fecal SCFA profiles in FRD dogs. Thus, it is unclear whether prebiotics and GAGs really improved intestinal environment in the GI tract of dogs. According to the introduction, GAGs have beneficial effects on intestinal barrier functions, microbiome, and fecal short chain fatty acid (SCFA). The authors, therefore, must analyze expression of tight junction proteins in the intestinal epithelial cells and gut microbiome in dogs that received prebiotics and GAGs or placebo.

Many thanks for this comment. As suggested by editors, we have not replied to this comment.

3. Page 7, Inclusion and exclusion criteria:

To justify the diagnosis and ensure the evenness of group 1 and 2, please provide hematology and serum biochemistry data including TLI, cPLI, and ACTH-stimulation test at visit 1 and compare them between two groups.

Many thanks for this comment. We have provided all raw data and statistical output sheets of the study in supplement 1.

4. Page 10, Assessment of relapse rate after switching back to normal diet: “Patients were considered to have relapsed if their CCECAI at visit 4 through 7 increased more than 30% from the previous visit.”

Please provide the scientific evidence for the relapse definition used in this study.

Many thanks for this comment. We have provided a reference for a previously published study where this cutoff has been successfully used.

Minor comments:

1. Page 1: FRD is not the abbreviation of “Food Responsive Chronic Enteropathy”.

Thank you for this comment. This has been corrected.

2. Page 5: Please check the references in the following sentences:

(a) While induction of remission is easily achieved with dietary treatment in dogs with FRD, relapses can occur when the dogs are switched back to their normal diet after the trial elimination diet (Ref).

(b) The effect of GAGs on enhancing the barrier function of the intestinal epithelium, has been shown to protect animals from translocation of pathogenic bacteria and LPS into the peripheral blood6,7,6,8.

3. The authors cited 15 references, but I can find reference no. more than 15 in the text. Please check the manuscript thoroughly.

Many thanks for this comment- all references have been checked and amended.

4. Page 13-14, Clinical severity and biochemical parameters:

Please describe reference ranges of biochemical parameters in the text.

Many thanks for this comment. We have provided all raw data, including biochemical parameters and reference ranges, in supplement 1.

---

## [Decision Letter · Decision Letter 1]

7 Sep 2021

Clinical Efficacy of Prebiotics and Glycosaminoglycans versus Placebo In Dogs with Food Responsive Enteropathy Receiving a Hydrolyzed Diet: A Pilot Study

PONE-D-21-11631R1

Dear Dr. Allenspach,

We’re pleased to inform you that your manuscript has been judged scientifically suitable for publication and will be formally accepted for publication once it meets all outstanding technical requirements.

Kind regards,

Mathilde Body-Malapel

Academic Editor

PLOS ONE

Additional Editor Comments (optional):

Reviewers' comments:

Reviewer's Responses to Questions

**Comments to the Author**

1. If the authors have adequately addressed your comments raised in a previous round of review and you feel that this manuscript is now acceptable for publication, you may indicate that here to bypass the “Comments to the Author” section, enter your conflict of interest statement in the “Confidential to Editor” section, and submit your "Accept" recommendation.

Reviewer #1: All comments have been addressed

2. Is the manuscript technically sound, and do the data support the conclusions?

Reviewer #1: Yes

3. Has the statistical analysis been performed appropriately and rigorously? 

Reviewer #1: Yes

4. Have the authors made all data underlying the findings in their manuscript fully available?

Reviewer #1: Yes

5. Is the manuscript presented in an intelligible fashion and written in standard English?

Reviewer #1: Yes

6. Review Comments to the Author

Reviewer #1: Thank you to the authors for addressing all my comments and concerns. Congratulations on the manuscript.

7. PLOS authors have the option to publish the peer review history of their article (what does this mean?). If published, this will include your full peer review and any attached files.

Reviewer #1: No

---

## [Editor Report · Acceptance letter]

7 Oct 2021

PONE-D-21-11631R1 

Clinical Efficacy of Prebiotics and Glycosaminoglycans *versus* Placebo In Dogs with Food Responsive Enteropathy Receiving a Hydrolyzed Diet: A Pilot Study 

Dear Dr. Allenspach:

I'm pleased to inform you that your manuscript has been deemed suitable for publication in PLOS ONE. Congratulations! Your manuscript is now with our production department. 

Kind regards, 

on behalf of

Dr. Mathilde Body-Malapel 

Academic Editor

PLOS ONE